# Preliminary Assessment of Bespoke (‘X-tec’) Silica Particles for IRS Applications

**DOI:** 10.3390/insects16090937

**Published:** 2025-09-05

**Authors:** Stephania Herodotou, Natalie Lissenden, Kevin Skinley, Derric Nimmo, Janneke Snetselaar, Amy Guy, Peter Myers, Svetlana Ryazanskaya

**Affiliations:** 1Innovative Vector Control Consortium, Liverpool School of Tropical Medicine, Pembroke Place, Liverpool L3 5QA, UK; stephanieherodotou@gmail.com (S.H.); natalie.lissenden@ivcc.com (N.L.); derric.nimmo@ivcc.com (D.N.); janneke.snetselaar@ivcc.com (J.S.); svetlana.ryazanskaya@landcent.nl (S.R.); 2Department of Chemistry, University of Liverpool, Liverpool L69 3BX, UK; kevins1@liverpool.ac.uk; 3Liverpool Insect Testing Establishment, Liverpool School of Tropical Medicine, Pembroke Place, Liverpool L3 5QA, UK; amy.guy@lstmed.ac.uk

**Keywords:** indoor residual spraying, IRS, bespoke silica, X-tec silica, clothianidin, IRS modelling

## Abstract

Indoor residual spraying (IRS) encounters significant challenges that affect product efficacy and cost-effectiveness, such as the varied surfaces on which a.i.’s are sprayed, absorption into those surfaces, suboptimal presentation of the insecticide on the surface, and the effect of surface pH on the insecticide’s longevity. Any technologies that can reduce the amount of insecticide required while maintaining comparative mortality could significantly reduce product costs and impact IRS cost-effectiveness. This study investigates the potential of bespoke X-tec silica particles as a unique carrier for insecticides. Clothianidin-loaded X-tec silica is sprayed at different application rates to see whether the insecticide content in the IRS formulation can be reduced while maintaining its residual biological activity. It is likely that, at a lower dose, less active ingredient is available for mosquito pickup, causing reduced mortality. Caution must be taken in extrapolating these results, as no prior confirmatory work on these silica particles exists. Nevertheless, the efficacy seen at lower doses suggests that silica could be used to improve the cost-effectiveness of IRS applications, and a cost–benefit analysis should be conducted.

## 1. Introduction

### 1.1. Indoor Residual Spraying (IRS) of Insecticides

Disease vector control mainly relies on two key interventions: insecticide-treated nets (ITNs) and indoor residual spraying (IRS). While ITN products predominately rely on the pyrethroid class of insecticides, IRS products employ a range of insecticidal classes, with alternative modes of action, which are effective against pyrethroid-resistant mosquitoes. IRS, however, is a comparatively expensive vector control intervention, with cost modelling of IRS campaigns suggesting that product costs comprise ~33% of the total costs [1]. New active ingredients (a.i.’s) in development are expected to be more expensive, potentially increasing product costs and, consequently, IRS campaign costs. IRS encounters significant challenges that affect product efficacy and cost-effectiveness, such as the varied surfaces on which a.i.’s are sprayed, absorption into those surfaces, suboptimal presentation of the insecticide on the surface, and the effect of surface pH on the insecticide’s longevity. Insecticides can also have significantly different physical–chemical properties, complicating the standardisation of IRS formulations. Therefore, any technologies that can reduce the amount of insecticide required while maintaining comparative mortality or that can simplify formulation development could significantly reduce product costs and impact IRS cost-effectiveness.

### 1.2. The Physical State of Insecticides and Biological Efficacy

In general, research has focused on the correlation between the total insecticide concentration of a product and its bioefficacy. However, the physical state of an insecticide, and how that may change over time, can play a significant role in understanding how insecticidal products perform in the lab and field.

The physical state of an insecticide can either be crystalline (solid with aligned molecular lattice bonds) or amorphous (solid with no orderly aligned lattice bonds). Several studies have shown that different insecticide crystal structures (polymorphs) can affect insecticidal activity, leading to higher insect mortality if the most efficacious polymorph is controlled and used [2,3,4,5,6,7]. When insecticides are in a crystalline form, the pickup by a target organism (i.e., a mosquito) is influenced by several factors, including crystal size and shape. However, in an amorphous form, it is believed that the insecticide material exhibits diffusion within the carrier substrate (including on the surface), potentially offering a larger surface area for pickup. So, insecticides presented in the amorphous state could be more efficacious than those presented in the crystalline state. Therefore, understanding which physical state of an insecticide gives the best efficacy and optimising formulations to maintain long-term stability are important factors to consider when developing an insecticidal product.

### 1.3. Silica as a Carrier for Insecticides

Silica has been used as a carrier for insecticides for decades [8]. It can offer the advantage of solid diffusion [9], a slow delivery mechanism that can reduce an insecticide’s environmental toxicity while increasing its bioefficacy against a target organism. However, the insecticide’s physical state and its presentation within or on a silica carrier are poorly understood and not controlled. It is possible that, by altering the parameters of the silica carrier, the crystallinity of an insecticide could be controlled by that carrier to hold it in an amorphous physical state. This is one of several ways that X-tec silica could enhance the bioefficacy of insecticides compared to standard IRS (Figure 1).

The critical crystallisation size of an insecticide refers to the minimum size at which insecticide particles can form stable crystals without redissolving. The critical crystallisation size can be understood by using mathematical and computer modelling [10,11]. These models can be applied to insecticides to provide a theoretical size at which the molecule can be maintained in an amorphous solid form.

To test this, a bespoke silica carrier (‘X-tec silica’) was developed with pores of a specific diameter and volume according to the specifications of the critical crystallisation size of an insecticide. To prove the concept of bespoke silica particles, clothianidin was chosen as an example insecticide, and X-tec silica was loaded with clothianidin. Clothianidin-loaded X-tec silica was sprayed at different application rates to see whether the insecticide content in the IRS formulation could be reduced while maintaining its residual biological activity. If similar or improved bioefficacy is achieved by reduced concentrations, or if the spray’s durability (longevity) is improved by the X-tec silica, this could result in cost savings compared to standard IRS formulations.

## 2. Materials and Methods

### 2.1. Molecular Computer Modelling of Silica Particles

Molecular computer modelling was used to simulate silica pores to determine the critical crystallisation size of clothianidin. The system used for the computer modelling was a parallel Linux cluster (Barkla, High Performance Computing resource, University of Liverpool, Liverpool, UK).

A preliminary analysis was conducted to demonstrate the capabilities of molecular computer modelling and understand how a functionalised silica surface can interact with water and functional groups. A molecular dynamics simulation was used to build an amorphous silica surface of a 1 nm thick slab in a 9 × 9 nm cell (Figure 2a). The surface was functionalised in the software by annealing and adding hydroxyl groups in a specific concentration. Then, a 6 nm pore was built to represent a real pore system (Figure 2b). The surface was further functionalised by adding C18 chain groups to reduce the hydrophilicity (Figure 2c). The mean maximum distance of the C18 from the silica surface was 1.4 nm, reducing the pore diameter to 3 nm. This amorphous silica surface model (Figure 2) was then used to illustrate the effect of absorbing clothianidin to the amorphous silica surface.

### 2.2. X-tec Silica Manufacture

X-tec silica was manufactured using a proprietary process and according to the specifications defined by the modelling analysis. The X-tec silica particles were then loaded with 10% clothianidin to produce the formulated product (hereafter referred to as ‘10% X-tec silica’).

### 2.3. Material Characterisation of 10% X-tec Silica

Material characterisation was conducted to measure the specifications of 10% X-tec silica to validate the manufacturing process and computer modelling using the following methods:

### 2.4. Carbon, Hydrogen, and Nitrogen (CHN) Analysis

A CHN analysis was carried out to confirm the loading of X-tec silica. Carbon, hydrogen, nitrogen, and sulphur were analysed using a ThermoFisher CHNS elemental analyser (https://www.thermofisher.com/). Dried and powdered samples were combusted in a tin sample crucible with a vanadium pentoxide catalyst in an oxygen environment. The resulting gas mixture containing N_2_, CO_2_, H_2_O, and SO_2_ flowed into the chromatographic column and was detected with a thermal conductivity detector (TCD).

### 2.5. Nitrogen Porosimetry

Nitrogen porosimetry was used to measure the surface area, pore diameter, and pore volume of the unloaded and 10% loaded silica particles, using nitrogen as the filling material. Apparent surface areas were measured by nitrogen adsorption at 77.3 K using a Micromeritics ASAP 2020 volumetric adsorption analyser (https://micromeritics.com/). Powder samples were degassed offline at 393 K for 12 h under a dynamic vacuum (10^−5^ bar). Before the adsorption test, the inert gas was removed using a high vacuum provided by the turbomolecular drag pump. The specific surface areas were evaluated using the Brunauer–Emmett–Teller (BET) model. The pore size distributions of the silica were obtained by fitting nonlocal density functional theory to the adsorption data.

### 2.6. Particle Size Measurement

To analyse the average particle size of the 10% X-tec silica, a powder multisizer technique (Beckman Coulter Multisizer 3; https://www.mybeckman.uk/) in volume mode was used to detect changes in electrical impedance as particles passed through a small aperture.

### 2.7. Scanning Electron Microscopy/Energy-Dispersive X-Ray Spectroscopy (SEM/EDS)

SEM/EDS was used as a quality control technique to assess whether the particles were damaged during manufacturing or formulation and as a qualitative technique to detect clothianidin on the X-tec silica. A powdered 10% X-tec silica sample was applied on a carbon tape and coated with chromium. Images of the sample’s surface were taken using an SEM (Tescan FIB SEM S8000G; https://tescan.com/). During the visual investigation of the surface, EDS (Oxford EDS system) was used for elemental analysis to confirm the origin of the features observed. Chlorine (Cl) and sulphur (S) were used to identify clothianidin (C_6_N_5_H_8_SO_2_Cl).

### 2.8. Powder X-Ray Diffraction (PXRD)

PXRD was carried out to confirm that the X-tec silica could maintain the amorphous physical state of clothianidin at 10% loading. PXRD patterns were recorded on a Bruker D8 Advance diffractometer (https://www.bruker.com/), with Cu Kα radiation and a voltage of 40 kV. Data were collected in the 2*θ* range of 2–40°, with steps of 0.02°.

### 2.9. Entomological Testing: Mosquito Rearing

The *Anopheles gambiae* s.s. Kisumu strain is reared and maintained in the Liverpool Insect Testing Establishment (LITE) insectaries at the Liverpool School of Tropical Medicine (LSTM) under controlled environmental conditions and a standardised blood feeding procedure (Williams et al., 2019) [12]. The strain was first colonised in Kenya in 1975 and was sourced from the Malaria Research and Reference Reagent Resource Centre. Adult mosquitoes are kept in 30 × 30 cm mesh cages at 27 ± 1 °C and 75% RH ± 5% with ad libitum access to a 10% glucose solution on cotton wool. The mosquitoes used in these experiments were non-blood-fed 2–5-day-old adult females.

### 2.10. Cone Bioassays

X-tec silica (10%) was mixed with water containing 2% *v*/*v* methanol and sprayed on glazed and unglazed tiles using an auto-load precision spray potter tower (Burkard Manufacturing Co., Ltd., Rickmansworth, UK) at three different application rates (30, 60, and 90 mg a.i./m^2^). After application, the tiles were kept for one week before testing to allow drying time. The tiles were tested at 1 week and 8 months post-spray application. The tiles were stored after spray application and between tests at 30 ± 2 °C and 80 ± 10% RH to replicate the temperatures and humidity typically encountered under field conditions.

The bioefficacy of the sprayed tiles was determined using a WHO cone bioassay [13]. Two to four tiles from each treatment group were tested at a time, exposing 10 ± 2 mosquitoes to each tile for 30 min. Mosquitoes exposed to the untreated tiles and unformulated X-tec silica tiles were used as controls. Knock-down was assessed at 60 min post-exposure, and mosquito mortality was recorded at 24, 48, 72, 96, and 120 h post-exposure. All tests were conducted at 27 ± 2 °C and 75 ± 10% relative humidity.

## 3. Results

### 3.1. Molecular Computer Modelling of Silica Particles

The molecular computer modelling suggested that the optimum specifications of X-tec silica particles to be used as carriers for clothianidin were 5 µm spheres manufactured by reaction-limited kinetics with 7–8 nm pores, a surface area of 200 m^2^/g, 2.0 silanols nm^2^, and a minimal of vicinal or geminal silanols (Figure 3).

### 3.2. X-tec Silica Manufacture

The manufactured silica particles had an average size of 5 µm spheres with 7 nm pores and a 170 m^2^/g surface area. The 10% X-tec silica had an average particle size of 5.64 µm.

### 3.3. Material Characterisation of 10% X-tec Silica

#### 3.3.1. CHN Analysis

The CHN analysis was carried out to confirm that the loading of X-tec silica was successful and occurred at the expected rate. The expected (calculated) value was 2.4% carbon, with the measured value being 3.9% carbon. The proximity of the estimated and measured values indicates that the silica was loaded with the target AI concentration.

#### 3.3.2. Nitrogen Porosimetry

Compared to the unloaded X-tec silica, the 10% X-tec silica showed significant reductions in the specific surface area (16% reduction) and mean pore volume (14% reduction) (Table 1). The mean pore diameter did not change after formulation.

#### 3.3.3. SEM/EDS

The SEM analysis found traces of clothianidin around the 10% X-tec silica. This was confirmed by detecting Cl and S (stoichiometrically equal, as it is on clothianidin) in the EDS (Figure 4). The silica particles maintained their shape without being damaged or altered during manufacture and formulation.

#### 3.3.4. PXRD Analysis

The PXRD analysis was conducted to confirm that the X-tec silica can maintain the amorphous physical state of clothianidin. The PXRD patterns showed no peaks, confirming that the sample contained an amorphous silica powder with no diffracting crystalline component (a representative example is shown in Figure 5).

### 3.4. Entomological Testing: Cone Bioassays

At 1 week post-application, on glazed tiles, all three application rates (30, 60, and 90 mg a.i./m^2^) of the 10% X-tec silica resulted in >80% mosquito mortality within 24 h (Figure 6 and Appendix A). Replicates were pooled following exposure, so, for this dataset, error bars could not be calculated. On unglazed tiles (Figure 7 and Appendix A), efficacy was lower; however, the two higher concentrations still surpassed >80% mortality (by 24 h for 90 mg a.i./m^2^, N = 41, and 48 h for 60 mg a.i./m^2^, N = 40). Mortality was <60% for the 30 mg a.i./m^2^ application rate at the final time point (120 h, N = 40). At 8 months post-spray application, on glazed tiles (Figure 8 and Appendix A), 100% mortality was reached within 24 h at 60 (N = 20) and 90 (N = 21) mg a.i./m^2^ application rates and within 48 h at 30 mg a.i./m^2^ (N = 20). On unglazed tiles (Figure 9 and Appendix A), 96 h mortality was not recorded; however, 100% mortality was reached within 72 h (90 mg a.i./m^2^, N = 18) and 120 h (60 mg a.i./m^2^, N = 19). For the lowest concentration (30 mg a.i./m^2^, N = 20), mortality only reached 25% within the 120 h period. For the NIRS-SB-0 X-tec silica control, mortality was 0% across all time points.

## 4. Discussion

This was not a confirmatory study but rather a proof of concept to test the feasibility of bespoke X-tec silica particles as a unique carrier for insecticide. Further studies on multiple active ingredients and specific IRS formulations are required before X-tec silica particles can be recommended for use in IRS.

The results, highlighted by the PXRD and SEM/EDS data, suggest that 10% X-tec silica particles were successfully manufactured to control the crystallinity of clothianidin and hold it in an amorphous physical state. The average particle size and SEM images show that the silica remained undamaged during the formulation process, with the CHN data indicating that the expected loading of clothianidin was successfully achieved in the silica batch. The surface area, measured by porosimetry, was significantly reduced in the 10% X-tec silica compared to the unformulated silica, indicating that the pores were filled up by the amorphous clothianidin. This study did not conduct a comparison on the efficacy of amorphous versus crystalline clothianidin, so it cannot determine which state is more efficacious. However, the ability to control the physical state of an insecticide could have far-reaching implications in cases where an insecticide is known to be more efficacious in a particular state, particularly if cost modelling shows that the costs of the bespoke silica are offset by the reduction in the amount of active ingredient needed to achieve the same or improved efficacy.

The 10% X-tec silica demonstrated high bioefficacy against the insecticide-susceptible Kisumu mosquito strain, with most dose/substrate combinations achieving 100% mortality within 120 h at 8 months post-spray application. The current clothianidin-based IRS product on the market specifies a target dose of 300 mg a.i./m^2^. Therefore, the demonstration of 100% efficacy at substantially reduced doses (60 and 90 mg a.i./m^2^) is positive. When calculating diagnostic concentrations, the dose that kills 100% of a susceptible strain is typically multiplied by 2. When doubling the susceptible dose that we observed to 120–180 a.i./m^2^), it is still significantly lower than the current market dose (300 mg a.i./m^2^). Even at a 10-fold reduced dose (30 mg a.i./m^2^), mortality was observed, although its efficacy was surface-dependent (1-week post-spray application: 100% on glazed tiles and 58% on unglazed tiles; 8-months post-spray application: 100% on glazed tiles and 25% on unglazed tiles). Although the X-tec silica was assumed to have reduced the absorption of the insecticide, the reduction in efficacy on the unglazed tiles at the lowest dose suggests that the effect of capillary action directly affects the bioavailability of the compound. It is likely that, at the lower dose, less active ingredient is available for mosquito pickup, causing reduced mortality. However, it is critical to understand that the presentation of any insecticide in any physical state will be impacted by the substrate on which it is sprayed. Secondly, caution must be taken in extrapolating these results, as no prior confirmatory work on these silica particles exists. Nevertheless, the efficacy seen at these lower doses suggests silica could be used to improve the cost-effectiveness of IRS applications, and a cost–benefit analysis should be conducted. Further testing should consider investigating the efficacy of the formulated silica on pyrethroid-resistant mosquito strains and exploring the efficacy of formulated silica with alternative active ingredients.

## 5. Conclusions

This study investigated the potential of bespoke X-tec silica particles as a unique carrier for insecticides. Preliminary findings from this study suggest that X-tec silica particles may enhance the effectiveness of IRS using clothianidin, maintaining its bioefficacy for up to eight months post-spray. However, further extensive research is needed to confirm this.

## Figures and Tables

**Figure 1 insects-16-00937-f001:**
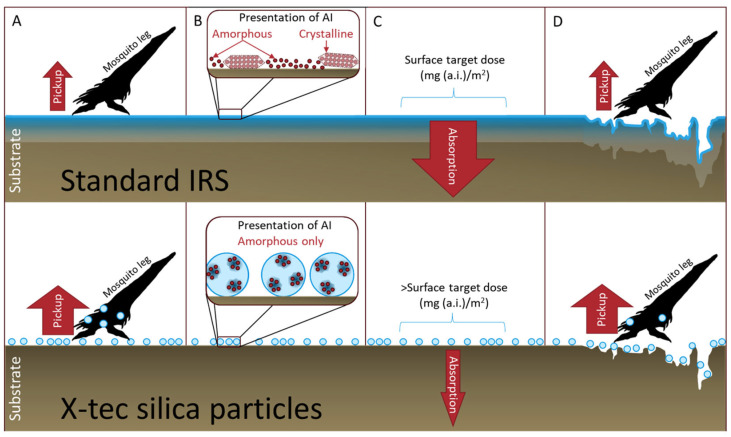
**Diagrammatic illustration of the potential benefits of using X-tec silica particles (bottom) to enhance insecticide bioefficacy compared to standard IRS (top).** (**A**) **Improved pickup**: The insecticide is loaded onto X-tec silica, which can be easily picked up by the mosquito leg compared to standard IRS. (**B**) **Physical state**: The X-tec silica maintains the insecticide in its amorphous state, the more efficacious form for most insecticides. In standard IRS, it is hard to maintain the amorphous state on the surface, so the insecticide can be amorphous or crystalline. (**C**): **Reduced absorption**: Many standard IRS formulations have challenges with absorption into porous surfaces (such as mud and wood), which reduces the surface concentration of the insecticide available to be picked up by the mosquito. X-tec silica should protect the insecticide, thereby eliminating this issue. (**D**): **Increased surface area**: Rough, irregular surfaces can affect pickup by reducing the surface area contact with the mosquito leg. The size of the X-tec silica particles can be adjusted to enhance pickup and reduce loss into the substrate surface.

**Figure 2 insects-16-00937-f002:**
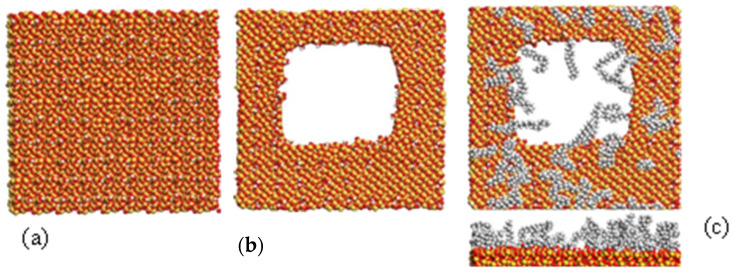
**Molecular computer modelling of amorphous silica surface**: (**a**) initial non-porous surface, (**b**) surface with 6 nm pores, (**c**) C18-functionalised surface top and side views.

**Figure 3 insects-16-00937-f003:**
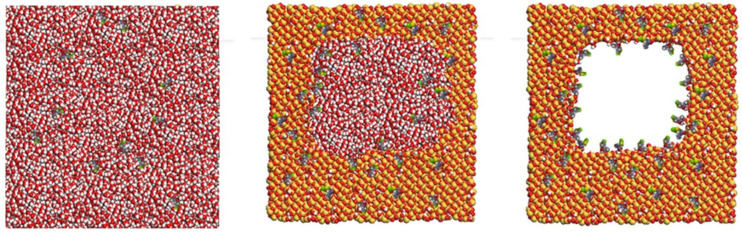
**Computer modelling of amorphous silica surface with 6 nm pores and clothianidin mixed on its surface:** (**left**) clothianidin molecules, (**middle**) amorphous silica surface mixed with clothianidin, and (**right**) the resulting silica surface with attached clothianidin molecules.

**Figure 4 insects-16-00937-f004:**
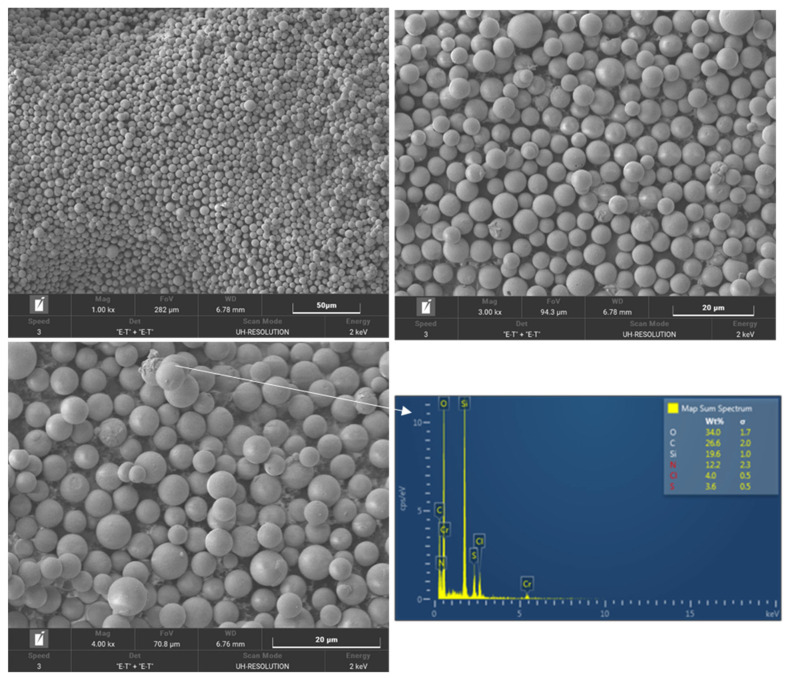
SEM/EDS analysis of the 10% X-tec silica.

**Figure 5 insects-16-00937-f005:**
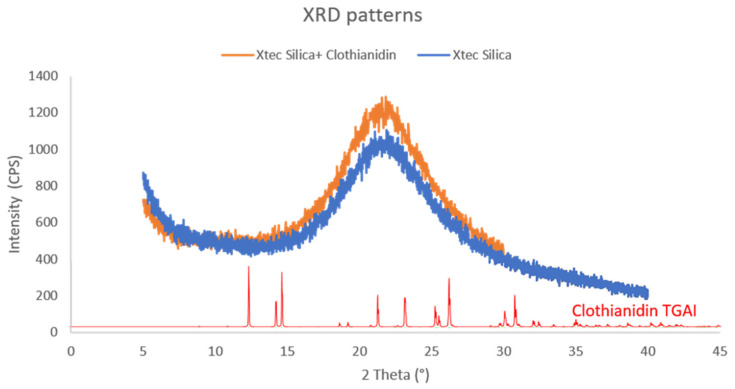
A representative pattern of the formulated X-tec silica. The PXRD pattern shows X-tec 5 µm silica formulated with 2 wt% clothianidin (orange), the pre-formulated blank X-tec silica (blue), and the reference PXRD pattern of crystalline clothianidin TGAI clothianidin (red). A PXRD pattern will show peaks when the material is crystalline, as X-rays will be reflected from the same angle when the lattice planes are parallel to each other (aligned), whereas, if the material is amorphous, the planes will have random alignment, and, hence, the X-rays will be scattered into different angles; therefore, no peaks will be recorded in the PXRD pattern.

**Figure 6 insects-16-00937-f006:**
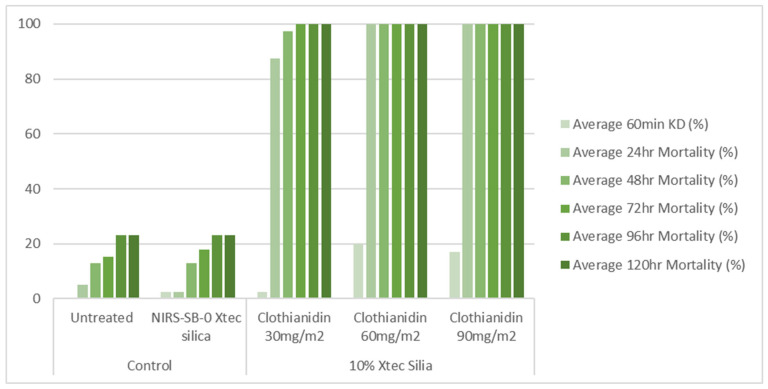
At 1 week post-spray application: 60 min knockdown (KD) and 24–120 h mortality of *An. gambiae* s.s. Kisumu exposed in cone bioassays to glazed tiles sprayed with 10% X-tec silica at three application rates (30, 60, and 90 mg a.i./m^2^), unformulated silica (NIRS-SB-0 X-tec silica), and an untreated control tile. Replicates were pooled; therefore, no error bars are shown.

**Figure 7 insects-16-00937-f007:**
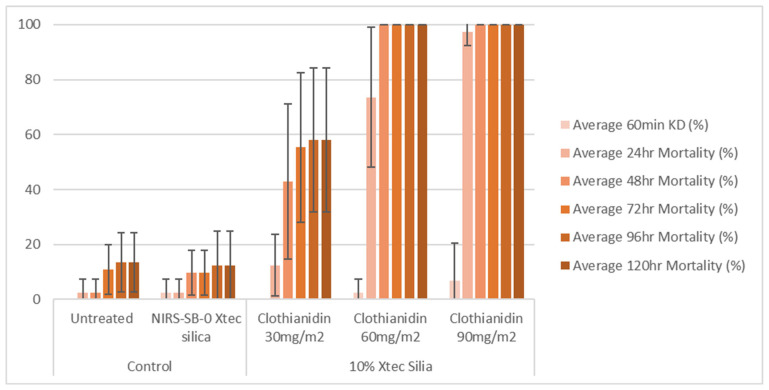
At 1 week post-spray application: 60 min knockdown (KD) and 24–120 h mortality of *An. gambiae* s.s. Kisumu exposed in cone bioassays to unglazed tiles sprayed with 10% X-tec silica at three application rates (30, 60, and 90 mg a.i./m^2^), unformulated silica (NIRS-SB-0 X-tec silica), and an untreated control tile. Error bars display the standard deviation of the sample.

**Figure 8 insects-16-00937-f008:**
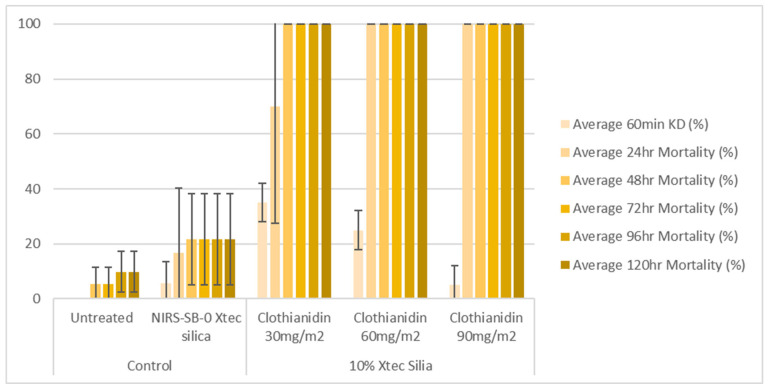
At 8 months post-spray application: 60 min knockdown (KD) and 24–120 h mortality of *An. gambiae* s.s. Kisumu exposed in cone bioassays to glazed tiles sprayed with 10% X-tec silica at three application rates (30, 60, and 90 mg a.i./m^2^), unformulated silica (NIRS-SB-0 X-tec silica), and an untreated control tile. Error bars display the standard deviation of the sample.

**Figure 9 insects-16-00937-f009:**
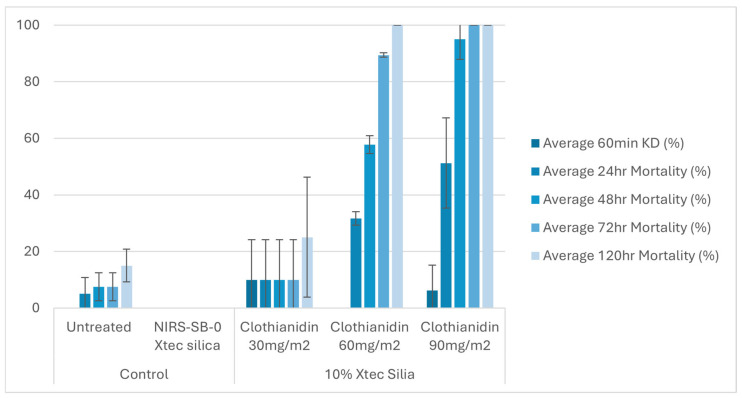
At 8 months post-spray application: 60 min knockdown (KD) and 24–120 h mortality (96 h mortality was not recorded) of *An. gambiae* s.s. Kisumu exposed in cone bioassays to unglazed tiles sprayed with 10% X-tec silica at three application rates (30, 60, and 90 mg a.i./m^2^), unformulated silica (NIRS-SB-0 X-tec silica), and an untreated control tile. Error bars display the standard deviation of the sample.

**Table 1 insects-16-00937-t001:** Summary of nitrogen porosimetry data detailing specific surface area (SSA), mean pore volume (MPV), and mean pore diameter (MPD) for unloaded and 10% clothianidin-loaded X-tec silica.

Name	SSA (m^2^/g)	MPV (cm^3^/g)	MPD (nm)
X-tec silica (unloaded)	167	0.42	7.3
X-tec silica (10% clothianidin)	140	0.36	6.9

## Data Availability

The original contributions presented in this study are included in the article/Appendix A. Further inquiries can be directed to the corresponding author.

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
