# Peer review of "Preliminary Assessment of Bespoke (‘X-tec’) Silica Particles for IRS Applications"

_insects, 2025, doi:10.3390/insects16090937_

Round 1
Reviewer 1 Report
Comments and Suggestions for Authors
Comments on each section of the manuscript
Abstract: The abstract says what the study is all about. However, it is too long with much details which relate to the method. It may be made short and concise. To me, key words seem to repeat themselves e.g. indoor residual spraying and IRS; bespoke silica and X-tec silica.
Introduction: The background is short and clear. Current knowledge related to research question is accurately described. However, Lines 122-130 seem to be a mixture of method and study aims. The study rationale and aims are not clearly defined and may need to be reformulated.
Materials and method: The study system was clearly described with the help of diagrammatic illustrations. Study outcomes were defined and outcome measures were clearly validated.
Results: The results were well represented supported by well-defined figures with well detailed figure legends.
Discussion: The authors discussed their findings well and identified limitations, research gaps and made recommendations for future studies.
Conclusion: The conclusion summarizes the study findings/ achievements well in one sentence.
Author Response
Please see our responses to the reviewers' comments in the attached ‘point by point response to reviewers' comments’. These have been included in the revised manuscript as tracked changes. We would like to extend our thanks to Reviewer 1, Reviewer 2, and the Editor for their support and comments on our manuscript.

Reviewer 2 Report
Comments and Suggestions for Authors
This article describes a carrier (bespoke silica particles) to enhance, by controlling the physical state, insecticide uptake specifically for an IRS based product with the active ingredient clothianidin.
See comments below (note this is a limited review and excludes reviewing the chemistry/ physical characteristics of the silica carrier)
Review:
Bespoke (‘X-tec) silica particles for IRS applications
Herodotou et al
Line 23: check spelling/format of X-tec
Figure 1: correct X-tec written format in figure
Line 107: should be tarsus (single leg) or leave out the word ‘the’ and make leg=legs
Line 172: please define ‘BET’ model
Line 198: italisize ‘ad libitum’
Line 202: write 10% in words or restructure sentence so it does not start with 10%
Line 207: please explain why tiles were stored at the specified temperature
Line 225: add the word ‘particles’ after silica
Line 226: use a hard space (control+shift+space) between the number and the unit to prevent the whole from separating and to keep it written in the same way as your other numbers and units
Lines 229, 234, 238: X-tec is written differently each time. Please choose one way of writing this and be consistent throughout the document
Line 255: confirm if 2wt.% Clothianidin is how it should be written
Lines 261-274 rather include the mortality (value/s) reached instead of >80% as it is usually well above 80%
Line 274: 30 mg a.i/m2 mortality reached 25% - specify the time-point at which 25% mortality was reached
Line 274: keep 'ai/m2' and 'N' together in brackets as done for the others
Figure 9: please remove -20 from the y axis of the graph. Why are there no bars for X-tec silica, provide an explanation please
Lines 314-315: perhaps specify that the dose that kills 100% of a susceptible strain is normally doubled to achieve the diagnostic concentration – so closer to 120/180mg a/i/m2 possibly (still significantly less than 300 mg a.i/m2
Line 321: missing a full stop after ‘compound’
Reference 13: WHO guideline for testing treated nets is not the correct one to use. These are normally 10 minute exposures
General comments: it would have been useful to include similar application rate simultaneously with a standard application (without X-tec silica) and a known application rate (at standard application method/carrier) that kills 100% of a susceptible population as a comparison and positive control respectively to the work conducted, this was mentioned in the discussion. It is recommended that these comparisons/ controls are included.
Sample sizes and number of replicates should be included in the results of the bioassays
Author Response

(The authors gave the same response as above.)
